# Complete functional analysis of type IV pilus components of a reemergent plant pathogen reveals neofunctionalization of paralog genes

**Marcus V. Merfa[1], Xinyu Zhu[2], Deepak Shantharaj[1], Laura M. Gomez[1], Eber Naranjo[1], Neha Potnis[1], Paul A. Cobine[2], Leonardo De La Fuente[1]***

**1** Department of Entomology and Plant Pathology, Auburn University, Auburn, Alabama, United States of America, **2** Department of Biological Sciences, Auburn University, Auburn, Alabama, United States of America

* lzd0005@auburn.edu

**Data Availability Statement:** All relevant data are within the manuscript and its Supporting Information files.

## Abstract

Type IV pilus (TFP) is a multifunctional bacterial structure involved in twitching motility, adhesion, biofilm formation, as well as natural competence. Here, by site-directed mutagenesis and functional analysis, we determined the phenotype conferred by each of the 38 genes known to be required for TFP biosynthesis and regulation in the reemergent plant pathogenic fastidious prokaryote *Xylella fastidiosa*. This pathogen infects > 650 plant species and causes devastating diseases worldwide in olives, grapes, blueberries, and almonds, among others. This xylem-limited, insect-transmitted pathogen lives constantly under flow conditions and therefore is highly dependent on TFP for host colonization. In addition, TFP-mediated natural transformation is a process that impacts genomic diversity and environmental fitness. Phenotypic characterization of the mutants showed that ten genes were essential for both movement and natural competence. Interestingly, seven sets of paralogs exist, and mutations showed opposing phenotypes, indicating evolutionary neofunctionalization of subunits within TFP. The minor pilin FimT3 was the only protein exclusively required for natural competence. By combining approaches of molecular microbiology, structural biology, and biochemistry, we determined that the minor pilin FimT3 (but not the other two FimT paralogs) is the DNA receptor in TFP of *X. fastidiosa* and constitutes an example of neofunctionalization. FimT3 is conserved among *X. fastidiosa* strains and binds DNA non-specifically via an electropositive surface identified by homolog modeling. This protein surface includes two arginine residues that were exchanged with alanine and shown to be involved in DNA binding. Among plant pathogens, *fimT3* was found in ~ 10% of the available genomes of the plant associated Xanthomonadaceae family, which are yet to be assessed for natural competence (besides *X. fastidiosa*). Overall, we highlight here the complex regulation of TFP in *X. fastidiosa*, providing a blueprint to understand TFP in other bacteria living under flow conditions.

**Funding:** We acknowledge funding from Alabama Agricultural Experiment Station (AAES) Hatch Program (L.D., P.A.C., N.P) and Auburn Internal Grant Program. The funders had no role in study design, data collection and analysis, decision to publish, or preparation of the manuscript.

**Competing interests:** The authors have declared that no competing interests exist.

## Author summary

*Xylella fastidiosa* is a bacterial plant pathogen that lives exclusively inside the xylem vasculature in plants and the food canal of insect vectors. Historically known to cause devastating diseases in the Americas in grapevines and other crops, in the last decade it has expanded to different parts of the world, notably the south of Italy where is killing ancient olive trees. We studied a bacterial structure, type IV pilus (TFP), that is important for cell movement and acquisition of DNA from the environment. During the functional characterization of each one of the 38 proteins involved in TFP formation and regulation, we noticed that multiple copies of the same genes (paralogs) had different functions, indicating that *X. fastidiosa* allocate several genes for a precise regulation of TFP. This highlights the key role of this structure in the pathogen life cycle. Moreover, we identified a specific protein that attaches DNA and is involved in the internalization of the genetic material used to recombine and generate bacterial variants that could have different virulence levels or host specificity. We hope that our complete phenotypic characterization of TFP genes will inspire others to further understand its role in vascular pathogens.

## Introduction

Type IV pili (TFP) are retractable hair-like proteinaceous cell appendages located at one or both poles of a bacterial cell [51]. In addition to playing fundamental roles in twitching motility and natural transformation, they also influence virulence, adhesion, and biofilm formation in many bacterial species [23, 24, 35]. During twitching motility, TFP extend, attach to surfaces, then retract pulling cells towards the point of attachment [35, 51]. TFP also plays a central part in the DNA-uptake machinery that enables natural transformation [23, 68]. Natural transformation is a horizontal gene transfer mechanism that allows bacterial cells to take up free DNA from the environment and incorporate it into their own genomes via homologous recombination (HR) [23, 37]. The current model of natural transformation in bacteria involves binding of double-stranded DNA to the tip of TFP, which upon retraction brings this molecule to the outer-membrane surface of the cell [24]. DNA is then transported to the periplasm in a proposed Brownian ratchet mechanism [24, 70]. Subsequently, DNA is translocated to the cytosol as a single strand, where recombination occurs if the incoming DNA shares homology with the genome of the recipient cell [23]. Alternatively, the incoming DNA is either metabolized or used as template to repair damaged DNA [69].

Natural transformation enables rapid evolution by generating genetic diversity and has been involved in spreading of antibiotic resistance, adaptation to new environments and emergence of pathogens [36, 37, 69, 77]. In addition to animal pathogens [36, 37], natural competence has been detected in two plant pathogens, *Ralstonia solanacearum* [7] and *Xylella fastidiosa* [43], both of which possess a very broad plant host range and are xylem-colonizers [20]. *X. fastidiosa* is a gram-negative fastidious prokaryote that infects and causes reemergent diseases in many economically important crops worldwide, such as grapevine, citrus, almond, peach, blueberry, and olive [13, 30, 62]. This pathogen lives constantly under flow conditions inside the xylem of infected plants and the foregut of xylem-feeding insect vectors [3]. The main mechanisms of virulence for *X. fastidiosa* are linked to its ability to move systemically within the xylem by TFP-mediated twitching motility, and formation of biofilm that disrupts sap flow [14].

*X. fastidiosa* has been classified into subspecies [57, 64, 65], and genome analyses revealed widespread HR events within and among subspecies [2, 11, 15, 33, 55, 56, 58, 59, 64, 75].

Moreover, experimental evidence showed that *X. fastidiosa* strains are able to recombine in vitro during coculture of either live or live and dead strains [39]. This process occurs under multiple experimental settings including batch and flow culture conditions with synthetic media or grapevine xylem sap from tolerant and susceptible varieties [38, 43, 44]. Finally, many of the genes that are undergoing recombination among strains in vitro or in nature, are annotated as having roles in virulence and bacterial fitness [59]. These observations led to the hypothesis that HR, a driver of genetic diversity, is responsible for environmental adaptation [59] and/or host switching/expansion. In fact, intersubspecific HR has been suggested to have a role in plant host expansion by generating strains that infect mulberry [57], citrus, coffee [56], blueberry and blackberry [58].

The assembly, function, and regulation of TFP require multiple molecular components. Minor pilins, which are present in much lower quantities than the major pilin, prime the assembly of TFP [53], and some species-specific minor pilins may promote additional functions of TFP, such as aggregation and adherence [27, 32]. The functional diversity in pilins is mediated by differences in the C-terminal region of these proteins, since all pilins (including major and minor pilins) share a highly conserved N-terminal region [27]. TFP is key to *X. fastidiosa* in both natural competence, which promotes HR and thus generates large genetic diversity [39, 59, 75], and twitching motility [14], which is its only mean of movement and fundamental for a xylem-limited bacterium living under flow conditions. Therefore, in this study we individually assessed the functional role of a comprehensive set of 38 TFP-related genes by site-directed mutagenesis. We identified ten core genes that were essential for both natural competence and movement of this plant pathogen. However, some components, mostly minor pilins, were only essential to one feature or the other, and interestingly most paralogous genes had opposite functions, suggesting neofunctionalization. Additionally, our study identified the FimT3 minor pilin as the DNA receptor of the *X. fastidiosa* pilus. This minor pilin is found among plant pathogens mainly in bacterial species of the Xanthomonadaceae family. Together, our data provide insight into how TFP paralogs evolve to perform specific functions within bacterial cells, and how different bacterial species use unique minor pilins to acquire exogenous DNA through natural competence.

## Results

### Deletion of TFP-related genes differentially affects natural competence and movement of *X. fastidiosa*

Individual deletion mutant strains were generated by site-directed mutagenesis via natural transformation to determine the functional roles of 38 TFP-related genes (S1 Table) [40]. Genes were chosen according to genome annotation and searches based on homology to counterparts encoded by *Pseudomonas aeruginosa* [9, 54]. To the best of our knowledge, these genes represent the entire set of TFP genes encoded by *X. fastidiosa* strain TemeculaL. Assessment of natural competence was conducted using the pAX1-Cm plasmid, which recombines into the neutral site 1 of the *X. fastidiosa* genome and inserts a chloramphenicol (Cm) resistance cassette [39, 50], whereas twitching motility was determined by measuring colony fringe widths on solid agar plates [25]. Although the knockout of genes studied here did not always modulate natural competence and twitching motility in the same manner, the correlation between these two traits among mutant strains was positive ($R^2 = 0.56$, $P<0.001$; S3 Table), indicating that they were usually increased or reduced in a similar manner upon deletion of TFP genes.

A set of ten genes were considered core TFP components as deletion of such genes abrogated both natural competence and movement of *X. fastidiosa* (Figs 1 and S1–S3). They

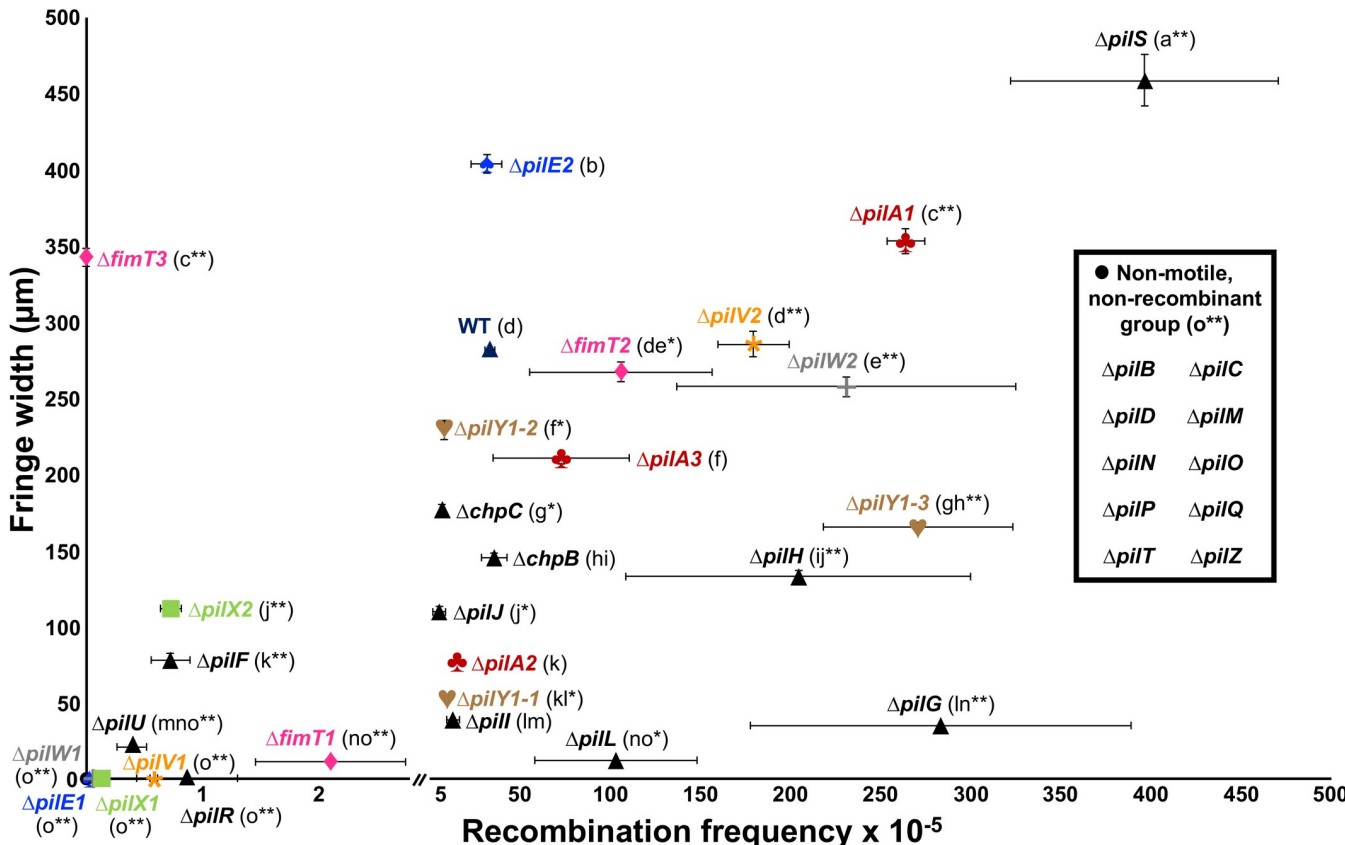

**Fig 1. Natural transformation and twitching motility phenotypes of *X. fastidiosa* mutant strains used in this study.** Quantification of natural transformation was performed by enumerating total viable and recombinant culturable cells and results are expressed as the ratio of recipient cells transformed (recombination frequency; values shown in the x-axis of the chart). Twitching motility was determined by spotting cells on agar plates and measuring the movement fringe width after 4 days of growth at 28°C (values shown in the y-axis of the chart). Results of mutant strains for paralogous genes are shown using the same color for text and symbol. The non-motile and non-recombinant mutants are shown as a black dot. The WT is represented as a dark blue triangle and all other mutants are represented as black triangles. Data represent means and standard errors. Different letters in parenthesis indicate significant difference in fringe width as analyzed by ANOVA followed by Tukey's HSD multiple comparisons of means ($P<0.05$; n = three to 14 independent replicates with eight to 48 internal replicates each). * and ** indicate significant difference ($P<0.05$ and $P<0.005$, respectively) of recombination frequency through natural competence in comparison to the WT as determined using Student's *t*-test (n = three to 21 independent replicates with two internal replicates each). The detection limit for recombination frequency was $10^{-7}$ and for twitching motility was 10 μm. Mutants below the detection limit are shown in the non-motile and non-recombinant group.

include the ATPases *pilB* and *pilT*, the prepilin peptidase *pilD*, the TFP assembly platform *pilC*, the whole alignment subcomplex *pilMNOP*, the secretin *pilQ* and the regulatory gene *pilZ*. Deletion of the sensor gene *pilS* of the PilRS two-component regulatory system (Figs 1 and S1–S3) significantly increased both natural competence and twitching motility, while deletion of the regulator *pilR* showed that this gene is essential for twitching motility but not for transformation. The minor pilin *fimT3* was the only gene found to be essential exclusively for natural competence (Figs 1 and S1). Curiously, this Δ*fimt3* strain showed significantly increased twitching motility than wild-type (WT) cells (Figs 1 and S2 and S3).

Seven sets of paralogs exist in the *X. fastidiosa* TFP cluster (PilA1-3, PilE1-2, PilV1-2, PilW1-2, PilX1-2, PilY1-1-3 and FimT1-3). Intriguingly, the knockout of paralogs presented varied (mostly opposing) phenotypes suggesting duplication and neofunctionalization of the genes. The minor pilins *pilE1V1W1X1*, which are encoded in the same operon (S2 Table), were essential for twitching motility but not for transformation (Figs 1 and S2 and S3), although these mutants did show significantly lower recombination frequencies than WT

(Figs 1 and S1). In comparison to the WT, Δ*pilE2* presented significant higher twitching motility and no changes in natural transformation, Δ*pilV2* had significant higher natural transformation with no altered movement, Δ*pilW2* showed significant higher natural transformation and significant lower movement, and Δ*pilX2* possessed both significant lower natural transformation and twitching motility (Figs 1 and S1–S3). Knockout of *pilA1* significantly increased both natural competence and twitching motility, whereas deletion of *pilA2* and *pilA3* significantly reduced twitching motility and did not alter natural transformation (Figs 1 and S1–S3). A strain with deletion of both *pilA1* and *pilA2* abolished twitching motility (Figs 1 and S2–S3), as described elsewhere [40]. Additionally, deletion of the minor pilin *fimT1* significantly decreased both natural competence and movement, while knockout of *fimT2* significantly increased natural competence and did not change twitching motility, and Δ*fimT3* lost natural competence and presented significantly higher movement than the WT (Figs 1 and S1–S3). As for *pilY1-1*, *pilY1-2* and *pilY1-3*, which are tip adhesins, their deletion significantly reduced twitching motility. However, Δ*pilY1-1* and Δ*pilY1-2* presented significant lower natural competence whilst Δ*pilY1-3* had significant higher natural competence than the WT (Figs 1 and S1–S3).

Lastly, deletion of the pilotin *pilF* and of the retraction ATPase *pilU* significantly reduced natural competence and movement. Knockout of the chemotaxis genes *pilGHIJL* and *chpBC* significantly reduced twitching. However, Δ*pilG*, Δ*pilH* and Δ*pilL* showed increased natural competence, while Δ*pilJ* and Δ*chpC* had lower natural competence, and Δ*pilI* and Δ*chpB* did not change this phenotype in comparison to the WT (Figs 1 and S1–S3).

## TFP-related genes differentially modulate other virulence traits of *X. fastidiosa*

We evaluated phenotypes such as growth, biofilm formation, cell aggregation (measured by settling rate), and virulence in planta of the individual TFP mutants. Growth rates were measured by change in $OD_{600nm}$ over time (S4 and S5 Figs). Since this growth analysis considered only turbidity that may be affected by cell attachment (biofilm) to surfaces and cell-to-cell aggregation, the number of culturable *X. fastidiosa* cells (measured as CFU/ml) was also evaluated. Only the non-recombinant and non-motile Δ*pilC* and Δ*pilP* mutants showed significant changes in the number of cells in comparison to WT, with both having significantly higher populations (S6 Fig). Biofilm formation is a critical aspect of virulence in *X. fastidiosa*. Individual deletion of *pilB*, *pilD*, *pilH*, *pilI*, *pilM*, *pilN*, *pilO*, *pilQ*, *pilR*, *pilT*, *pilX1* and *fimT1* significantly increased biofilm formation, whereas Δ*pilA3* and Δ*fimT3* formed significant less biofilm than WT (S7 Fig). Planktonic growth was significantly increased for strains with individual deletion of *pilA1*, *pilC*, *pilE1*, *pilF*, *pilM*, *pilN*, *pilO*, *pilP*, *pilQ*, *pilR*, *pilV1*, *pilW1*, *pilW2*, *pilX1*, *pilA1pilA2*, *pilY1-1*, *fimT3* and *chpC*, while individual knockout of *pilD*, *pilT* and *pilY1-3* significantly decreased planktonic growth (S8 Fig). Moreover, only Δ*pilD*, Δ*pilT* and Δ*pilX2* showed significant higher settling rates, with other mutants not differing significantly from the WT (S9 Fig). Biofilm formation and settling rate, as well as planktonic growth and growth rate were positively correlated (S3 Table). Meanwhile, biofilm formation and twitching motility/growth rate, settling rate and planktonic growth/growth rate, and growth rate and total viable *X. fastidiosa* had significant negative correlations (S3 Table). No other phenotype presented a significant correlation to recombination frequency besides twitching motility.

Virulence of a subset of the *X. fastidiosa* mutant strains was assessed by inoculating *Nicotiana tabacum* L. cv. Petite Havana SR1 plants and scoring disease incidence and severity. Mutants chosen for analysis had different twitching motility phenotypes, such as higher movement (Δ*pilA1* and Δ*pilS*), lower movement (Δ*pilA2*) and non-motility (Δ*pilA1pilA2*, Δ*pilQ* and

ΔpilR). Additionally, mutants in the DNA-binding *fimT3* (see below) and the paralogs *fimT1* and *fimT2* were tested and no significant change in virulence was found compared to WT. WT-inoculated plants presented the highest disease severity over the time course of assays (Fig 2A), with all plants inoculated with the mutant strains having significantly lower AUDPC (Area Under the Disease Progress Curve) values, except plants inoculated with ΔpilA2 (Fig 2B). All inoculated plants reached 100% disease incidence at the end of assays, except plants inoculated with ΔpilA1, which had 89% disease incidence (±11%, standard error). However, only WT-inoculated plants consistently presented severe symptoms of leaf scorch, while plants inoculated with the mutant strains mostly presented mild symptoms (Fig 2D). In addition, when evaluating the population of *X. fastidiosa* in planta by qPCR, only the ΔpilA1pilA2 double mutant showed a defect in colonizing the top leaf of the plant host, and the ΔpilQ and ΔpilR mutants presented significant higher population in the basal leaf of plant hosts (Fig 2C). Together, results demonstrate that the deletion of analyzed genes significantly reduced symptoms development in infected tobacco plants (except for deletion of *pilA2*), which was not explained by bacterial distribution, since most strains could effectively colonize the entirety of plants (except ΔpilA1pilA2).

## The minor pilin FimT3 is the DNA receptor of the *X. fastidiosa* type IV pilus

The observation of the minor pilin FimT3 being the only TFP component that is essential exclusively for natural transformation (Figs 1 and S1–S3) led us to hypothesize that this component is the DNA receptor of the TFP of *X. fastidiosa*. We first analyzed the piliation phenotype of ΔfimT3 in comparison to WT by transmission electron microscopy (TEM). Both presented long TFP as well as short type I pili, with no apparent changes in the piliation of *X. fastidiosa* cells upon deletion of *fimT3* (Fig 3A). Next, the ability of these cells to take up DNA from the extracellular environment was evaluated by exposing them to Cy-3-labeled pAX1-Cm plasmid and observing under a fluorescence microscope after a DNase I treatment to remove extracellular DNA. Upon exposure to the fluorescently labeled DNA fragment, WT cells imported DNA allowing for protection from exogenous DNase I, but no fluorescent DNA foci were present in ΔfimT3 cells (Fig 3B). DNA foci were mostly observed at one pole of the cells, which coincides to the described phenotype of *X. fastidiosa* that forms TFP at only one of its poles [52]. The DNA-uptake ability of the mutant strains of other minor pilins that decreased the natural transformation ability of *X. fastidiosa* upon deletion was also evaluated for comparison. These included ΔpilE1, ΔpilV1, ΔpilW1, ΔpilX1, ΔpilX2, ΔfimT1 and ΔfimT2 (Figs 1 and S1). The ΔfimT2 mutant strain was included because this gene is a paralog of *fimT3*, even though its deletion increased the natural competence of *X. fastidiosa* (Figs 1 and S1). All analyzed mutant strains maintained the DNA-uptake (competence) phenotype (S10 Fig), but at different levels (S11 Fig). WT cells presented the highest percentage of cells with DNA foci, with ΔpilE1 having the lowest proportion of cells with DNA acquisition (S11 Fig).

To determine the DNA-binding ability of FimT3, and compare it to its paralogs FimT1 and FimT2, the soluble portion of these proteins (named FimT3s, FimT1s and FimT2s, respectively), which lacked the conserved N-terminal hydrophobic α-helices of pilins [27], were amplified and cloned for expression. These proteins were then purified and assayed for DNA binding by agarose electrophoretic mobility shift assay (EMSA). Results showed FimT3s was the only protein capable of binding DNA, since no shift was observed with FimT1s and FimT2s when using as much as 10 μM of each protein (Fig 3C). Moreover, FimT3s was able to bind to different DNA sequences, independent of homology to the *X. fastidiosa* TemeculaL genome, indicating that binding is not sequence-specific (S12 Fig). Collectively, *fimT3* being

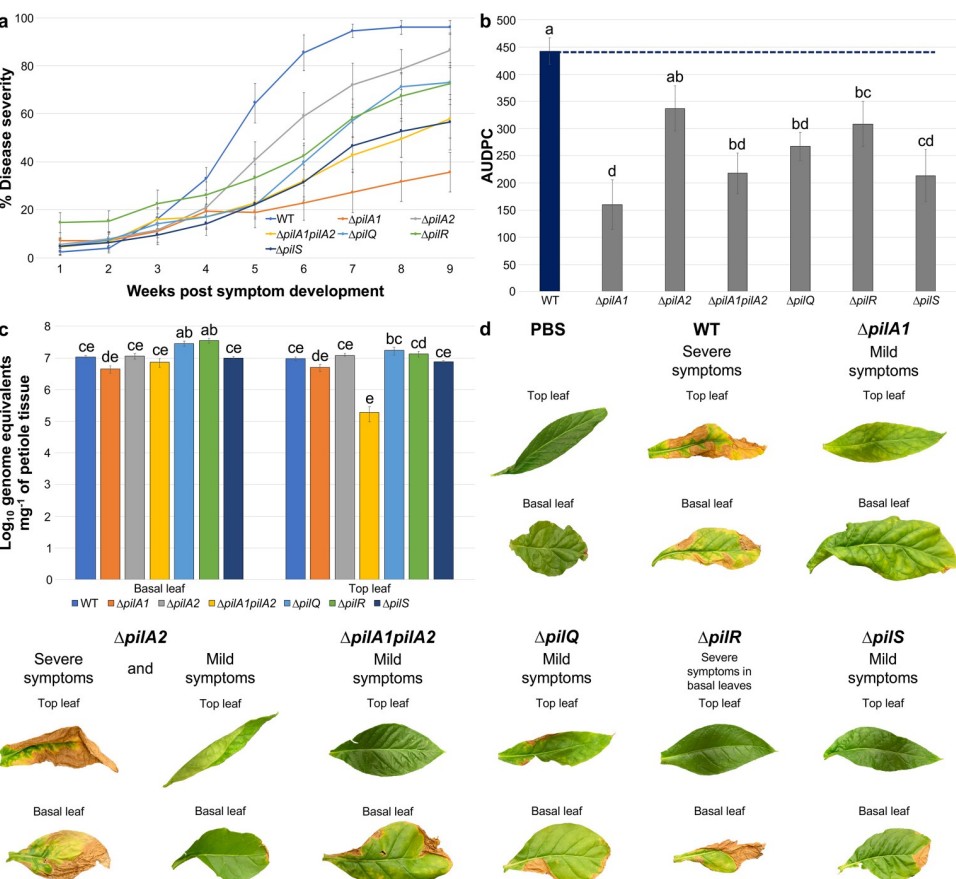

**Fig 2. Deletion of *pil* genes negatively affects virulence of *X. fastidiosa*. a** Disease severity progression over time in inoculated tobacco plants. *X. fastidiosa* WT and mutant strains were inoculated into *Nicotiana tabacum* L. cv. Petite Havana SR1 plants (PBS mock inoculation used as control). Mutant strains chosen for inoculation presented different twitching motility phenotypes, including higher twitching motility (ΔpilA1 and ΔpilS), lower twitching motility (ΔpilA2) and non-motility (ΔpilA1pilA2, ΔpilQ and ΔpilR). Leaf scorch symptoms were recorded for measurements of disease incidence and severity once a week during nine weeks after appearance of the first disease symptoms. At the final time point of evaluation, disease incidence in all inoculated plants reached 100%, except for plants inoculated with ΔpilA1, which reached 88.88% disease incidence (±11.11%, standard error). On the other hand, disease severity reached 96% in the WT, 35% in ΔpilA1, 86% in ΔpilA2, 57% in ΔpilA1pilA2, 73% in ΔpilQ, 72% in ΔpilR and 56% in ΔpilS. Data represent means and standard errors from two independent experiments (n = seven to ten plants in each independent experiment). **b** Mean AUDPC per treatment group. AUDPC was calculated using data from disease severity over nine weeks after first disease symptom appearance. WT is highlighted in blue, and the dashed blue line indicates the mean value of AUDPC for WT-inoculated plants. AUDPC was significantly lower for plants inoculated with all mutant strains in comparison to WT-inoculated plants, except for ΔpilA2. Data represent means and standard errors. Different letters on top of bars indicate significant difference as analyzed by ANOVA followed by Tukey's HSD multiple comparisons of means ($P<0.05$; n = two independent experiments with seven to ten plants each). **c** *X. fastidiosa* in planta population determined by qPCR at the last time point of evaluation. The population of WT and mutant strains throughout inoculated plants was calculated using petioles of basal and top leaves. ΔpilA1pilA2 cells showed a defect in colonizing the top leaf of plant hosts, while ΔpilQ and ΔpilR showed a significant higher population in the basal leaf of infected plants. This shows that mutant strains still move inside the xylem of plant hosts, but absence of deleted genes impairs full symptom development. Data represent means and standard errors. Different letters on top of bars indicate significant difference as analyzed by ANOVA followed by Tukey's HSD multiple comparisons of means ($P<0.05$; n = two independent experiments with three leaf replicates each). **d** Figure panel with representative pictures of leaf scorch symptoms in WT- and mutant strains-inoculated plants (top and basal leaves), as well as control plants (PBS mock inoculation). Only WT-inoculated plants consistently presented severe leaf scorch symptoms. Similar events were captured in two independent experiments.

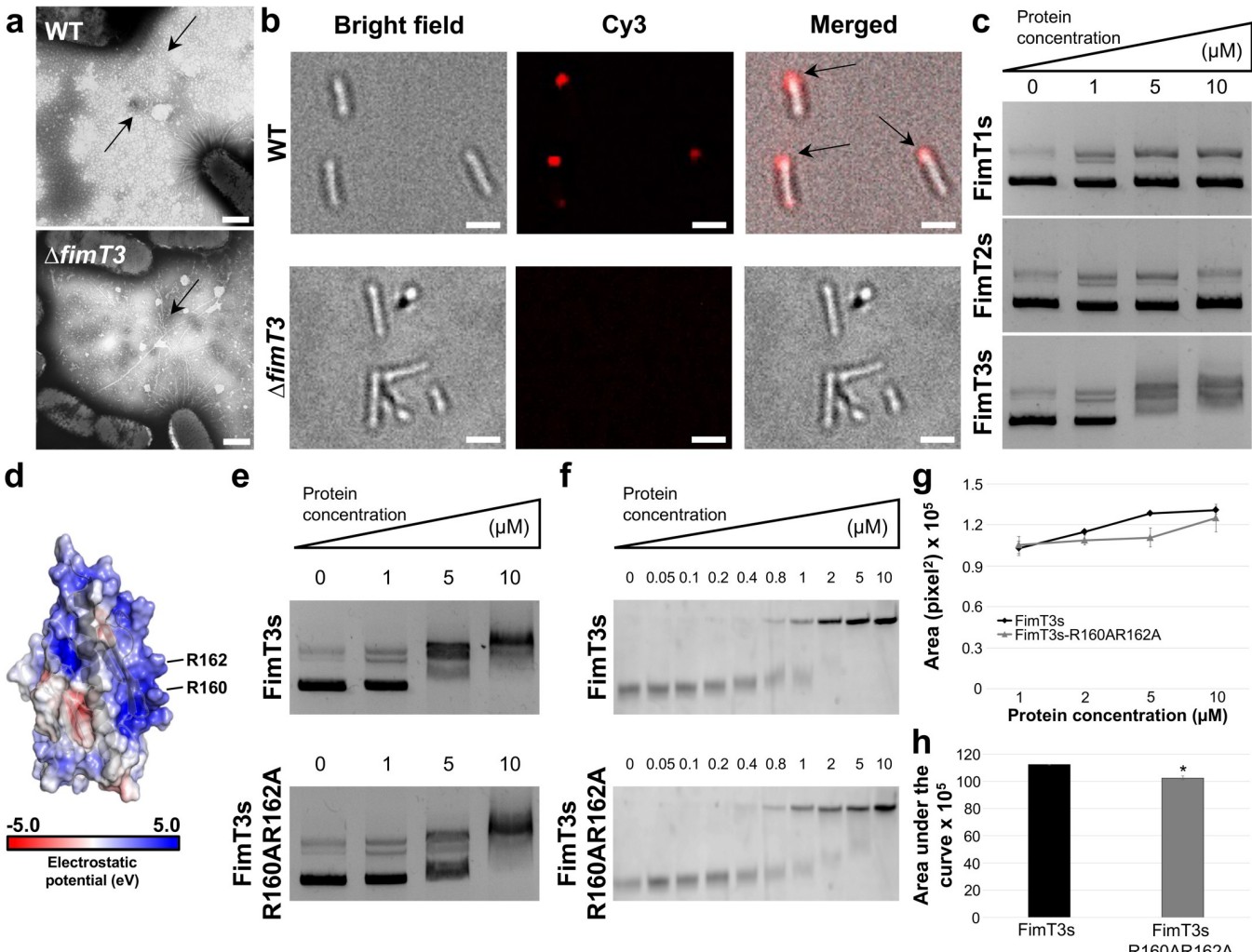

**Fig 3. FimT3 DNA binding activity. a** Transmission electron microscopy micrographs of pilus formation by *X. fastidiosa* WT and Δ*fimT3* cells. Arrows are pointing to TFP. Similar events were captured in two independent experiments. Images were captured at 31,500× magnification. Scale bars, 500 nm. **b** Uptake of Cy-3-labeled pAX1-Cm plasmid into a DNase I resistant state by *X. fastidiosa* cells observed using a fluorescence microscope. The images correspond to the bright field (left), Cy-3 channel (center) and merged images (right). In merged images, arrows are pointing to fluorescent DNA foci at the cell poles. Similar events were captured in three independent experiments. Images were captured at 100× magnification. Scale bars, 1.5 μm. **c** DNA-binding ability of purified FimT1s, FimT2s and FimT3s assessed by agarose EMSA. Similar events were captured in three independent experiments. **d** Surface electrostatics representation of FimT3 merged with its ribbon representation. The arginine amino acid residues (R160 and R162) that were mutated for functional studies are highlighted in the figure. The electrostatic potential (eV) color scale is shown in the figure. **e** DNA-binding ability of purified FimT3s and FimT3s-R160AR162A assessed by agarose EMSA. Similar events were captured in three independent experiments. **f** Titration of the DNA binding activity of FimT3s and FimT3s-R160AR162A by native acrylamide EMSA. Similar events were captured in two independent experiments. **g** Densitometry analysis of the DNA binding activity of FimT3s and FimT3s-R160AR162A in acrylamide EMSA. The fluorescent intensity of the shifted DNA bands of Cy-3-labeled Km resistance cassette presented in **f** was measured using ImageJ. The fluorescent intensity of shifted bands when treated only with 1, 2, 5, and 10 μM of each protein was measured, since they presented higher shifts in electrophoretic mobility within this protein concentration range. **h** Area under the fluorescence curve. The area under the fluorescence curves in **g** was calculated to quantify the DNA-binding affinity of FimT3s and FimT3s-R160AR162A. Statistical significance was determined using Student's *t*-test (* indicates *P*<0.05 in comparison to the wild-type protein; n = two independent replicates). Multiple bands of the pAX1-Cm plasmid observed in **c** and **e** are due to the multiple forms of the plasmid after purification, with each form displaying different sizes in agarose gels.

the only gene essential exclusively for natural competence, the ability of FimT3 to bind to DNA and the loss of DNA acquisition from the extracellular environment by cells upon its deletion, demonstrate that this minor pilin is the DNA receptor of the *X. fastidiosa* TFP.

To further characterize FimT3, we searched for specific amino acid residues involved in DNA-binding. The binding of DNA to TFP via a minor pilin was demonstrated in ComP of

*Neisseria*, ComZ of *Thermus thermophilus*, and VC0858 of *Vibrio cholerae* [12, 24, 61]. Amino acids alignments revealed that FimT3 and ComP share only 28% identity (64% similarity) over 20% of query sequence, while FimT3 and ComZ have no significant identity. Alignment of FimT3 with VC0858 of *Vibrio cholerae* showed 35% identity and 62% similarity over 18% of query sequence. In spite of the low overall homology the sequence alignments showed that an arginine residue at position 162 of FimT3 aligned with a lysine of ComP (K108) that was previously demonstrated to be essential for its DNA-binding ability [12] (S13A Fig). Additionally, an arginine residue at position 160 of FimT3 aligned with an arginine of VC0858 (R168) that was also important for the DNA-binding activity of the *V. cholerae* pilus [24] (S13B Fig). These two arginine residues of FimT3 (R160 and R162) were predicted in silico to have the highest probabilities of binding DNA among the residues of this protein (S4 Table). Moreover, alignment of FimT3 with a FimT ortholog that is essential for natural competence of *Acinetobacter baylyi* [46] (28% identity and 46% similarity over 91% of query sequence) showed alignment of R162 of FimT3 to an arginine of FimT from *A. baylyi*. Likewise, FimT3 shares 27% identity (48% similarity) over 80% of query sequence with a recently described DNA-binding FimT ortholog of *Legionella pneumophila*, in which the aligned R160 and R162 have been demonstrated as important to bind DNA [8] (S13C Fig).

Analysis of a structural model produced by the Phyre2 algorithm, showed that R160 and R162 of FimT3 are located within an electropositive surface and could form a pocket to bind DNA (Fig 3D). Thus, to assess the role of these arginine residues in the DNA-binding ability of FimT3, we generated a FimT3s variant by site-directed mutagenesis in which both arginine residues were replaced by alanine residues (FimT3s-R160AR162A). Agarose EMSAs revealed that WT-FimT3s had a higher affinity for DNA than FimT3s-R160AR162A based on the amount of protein needed to shift 50% of the input DNA (Fig 3E). Similar results were obtained when using the more sensitive acrylamide EMSA and the kanamycin (Km) resistance cassette sequence as target DNA (previously determined to be bound by FimT3s in S12 Fig). An abrupt shift was detected when using concentrations of FimT3s ranging between 1 and 5 μM, whereas a much more subtle and gradual shift was observed for samples treated with FimT3s-R160AR162A within the same concentration range (Fig 3F). We then performed a densitometry analysis of the shifted bands treated with concentrations ranging between 1 and 10 μM of each protein (Fig 3G) and calculated the area under the curve to quantify the DNA-binding affinity of FimT3s and FimT3s-R160AR162A. While the DNA-binding affinity of FimT3s-R160AR162A was lower than that of FimT3s (Fig 3H), the mutations did not completely abolish DNA-binding ability suggesting other residues are contributing to this activity. Nonetheless, these results show that R160 and R162 are important for DNA-binding activity of FimT3.

## FimT3 is widely conserved within *X. fastidiosa* strains and, among plant pathogens, is encoded by other members of the Xanthomonadaceae family

Previously, DNA-binding homologues of FimT3 have been found in many representative species of γ-Proteobacteria, including the Xanthomonadaceae family to which *X. fastidiosa* belongs [8]. We performed searches using the Conditional Reciprocal Best BLAST (CRB-BLAST) [4], as well as phylogenetic analyses, to further explore the distribution of the *fimT* paralogs within this family. All three paralogs of *fimT* are encoded by *X. fastidiosa*, *X. taiwanensis*, many clade II *Xanthomonas* spp. [34] and few *Pseudoxanthomonas* spp. (Fig 4). Curiously, early branching clade I *Xanthomonas* spp. only encode *fimT1* and *fimT3* (Fig 4). Specifically, *fimT3* was present in 1,416 out of 3,001 genomes of the Xanthomonadaceae (47%), the majority of them belonging to species within the genus *Xanthomonas*. In the *fimT3*

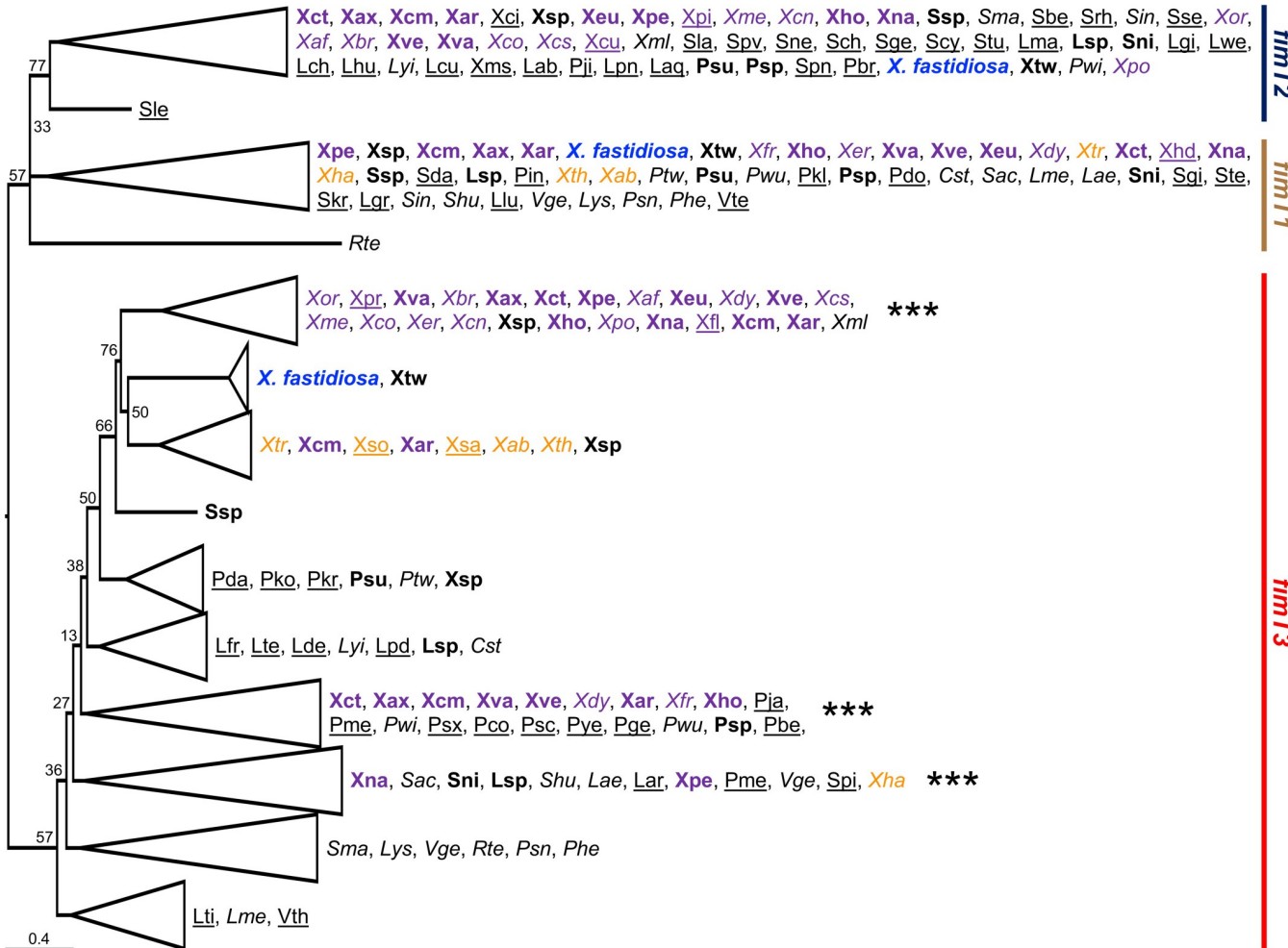

**Fig 4. Phylogenetic tree based on nucleotide sequences of the three *fimT* paralogs from Xanthomonadaceae bacterial strains with whole-genome sequences available.** The phylogenetic tree was built using the Maximum-likelihood method and visualized using FigTree. Branches with bootstrap values below 70% were collapsed, except where indicated. This was performed to keep conciseness of the figure while indicating the different clusters for *fimT1*, *fimT2* and *fimT3*. *X. fastidiosa* is highlighted in blue within the tree. When known, clade I *Xanthomonas* spp. are highlighted in orange, and clade II *Xanthomonas* spp. are highlighted in purple. Bacterial species encoding all three *fimT* paralogs are indicated in bold, while those encoding two paralogs are shown in italics and those encoding only one paralog are underlined within the tree. The order of appearance of bacterial species in collapsed branches follows the original order of appearance within the tree. For representation purposes, branches are highlighted by color according to each *fimT* paralog. Brown: *fimT1*; Dark blue: *fimT2*; Red: *fimT3*. \*\*\* Indicates clusters within the *fimT3* clade which contain few FimT3 sequences with no GRxR motif (see S5 Table and S14 Fig). Abbreviations are the following. ***Coralloluteibacterium:*** Cst–*C. stylophorae*. ***Luteimonas:*** Lab–*L. abyssi*; Lae–*L. aestuarii*; Laq–*L. aquatica*; Lar–*L. arsenica*; Lch–*L. chenhongjianii*; Lcu–*L. cucumeris*; Lde–*L. deserti*; Lfr–*L. fraxinea*; Lgi–*L. gilva*; Lgr–*L. granuli*; Lhu–*L. huabeiensis*; Llu–*L. lumbrici*; Lma–*L. marina*; Lme–*L. mephitis*; Lpd–*L. padinae*; Lpn–*L. panaciterrae*; Lsp–*L.* sp.; Lte–*L. terrae*; Lti–*L. terricola*; Lwe–*L. wenzhouensis*; Lyi–*L. yindakuii*. ***Lysobacter:*** Lys–*L.* sp. ***Pseudoxanthomonas:*** Pbe–*P. beigongshangi*; Pbr–*P. broegbernensis*; Pco–*P. composti*; Pda–*P. daejeonensis*; Pdo–*P. dokdonensis*; Pge–*P. gei*; Phe–*P. helianthi*; Pin–*P. indica*; Pja–*P. japonensis*; Pji–*P. jiangjuensis*; Pkl–*P. kalamensis*; Pko–*P. kaohsiungensis*; Pkr–*P. koreensis*; Pme–*P. mexicana*; Psc–*P. sacheonensis*; Psn–*P. sangjuensis*; Psp–*P.* sp.; Psx–*P. spadix*; Psu–*P. suwonensis*; Ptw–*P. taiwanensis*; Pwi–*P. winnipegensis*; Pwu–*P. wuyuanensis*; Pye–*P. yeongjuensis*. ***Rehaibacterium:*** Rte–*R. terrae*. ***Silanimonas:*** Sle–*S. lenta*. ***Stenotrophomonas:*** Sac–*S. acidaminiphila*; Sbe–*S. bentonitica*; Sch–*S. chelatiphaga*; Scy–*S. cyclobalanopsidis*; Sda–*S. daejeonensis*; Sge–*S. geniculata*; Sgi–*S. ginsengisoli*; Shu–*S. humi*; Sin–*S. indicatrix*; Skr–*S. koreensis*; Sla–*S. lactitubi*; Sma–*S. maltophilia*; Sne–*S. nematodicola*; Sni–*S. nitritireducens*; Spn–*S. panacihumi*; Spv–*S. pavanii*; Spi–*S. pictorum*; Srh–*S. rhizophila*; Sse–*S. sepilia*; Ssp–*S.* sp; Ste–*S. terrae*; Stu–*S. tumulicola*. ***Vulcaniibacterium:*** Vge–*V. gelatinicum*; Vte–*V. tengchongense*; Vth–*V. thermophilum*. ***Xanthomonas:*** Xab–*X. albilineans*; Xaf–*X. alfalfae*; Xar–*X. arboricola*; Xax–*X. axonopodis*; Xbr–*X. bromi*; Xcm–*X. campestris*; Xcn–*X. cannabis*; Xcs–*X. cassavae*; Xci–*X. cissicola*; Xct–*X. citri*; Xco–*X. codiaei*; Xcu–*X. curcubitae*; Xdy–*X. dyei*; Xer–*X. euroxanthea*; Xeu–*X. euvesicatoria*; Xfl–*X. floridensis*; Xfr–*X. fragariae*; Xho–*X. hortorum*; Xha–*X. hyacinthi*; Xhd–*X. hydrangea*; Xml–*X. maliensis*; Xms–*X. massiliensis*; Xme–*X. melonis*; Xna–*X. nasturtii*; Xor–*X. oryzae*; Xpe–*X. perforans*; Xpi–*X. pisi*; Xpo–*X. populi*; Xpr–*X. prunicola*; Xsa–*X. sacchari*; Xso–*X. sontii*; Xsp–*X.* sp.; Xth–*X. theicola*; Xtr–*X. translucens*; Xva–*X. vasicola*; Xve–*X. vesicatoria*. ***Xylella:*** Xtw–*X. taiwanensis*.

clade, *X. fastidiosa* strains grouped together, including *X. taiwanensis*, with a similar trend being observed for *Xanthomonas* spp. strains (Fig 4). However, clade I and clade II *Xanthomonas* spp. and other Xanthomonadaceae species clustered mostly separated (Fig 4). The alignment of the different FimT3 sequences encoded by *X. fastidiosa* strains showed a maximum of six nonconservative substitutions within strains belonging to the subspecies *pauca*, which encode FimT3 that are 13 amino acids shorter than other strains (S14A Fig). Alignment of representative FimT3 sequences among strains belonging to the Xanthomonadaceae family revealed that the arginine residues at positions 160 and 162 of FimT3 from *X. fastidiosa* strain TemeculaL were conserved in most orthologous sequences of this protein (S14B Fig). Additionally, R160 was always preceded by a glycine residue, constituting a highly conserved GRxR motif (S14B Fig), as seen elsewhere [8]. Although amino acid identities ranged from 95.60% to 100% within *X. fastidiosa*, and from 15.5% to 80% compared to other species, only 41 out of 1,416 FimT3 sequences encoded by the Xanthomonadaceae do not carry the GRxR motif (S5 Table). Overall, results indicate that FimT3 is encoded by representative plant pathogens of the Xanthomonadaceae and that R160 and R162 are highly conserved within FimT3 sequences.

## Discussion

In this study we defined the roles of individual subunits involved in regulation, assembly, and functioning of the *X. fastidiosa* TFP on natural competence, twitching motility, adhesion, and biofilm (Fig 5). To our knowledge, our study comprises the largest simultaneous functional description of a comprehensive set of TFP-related genes. By performing site-directed mutagenesis of all TFP genes followed by functional studies, we defined a set of ten core genes (*pilBCDMNOPQTZ*) that were essential for natural competence and movement. Most of these proteins are inner- or outer-membrane associated, confirming the important role of membrane anchorage for functionality of TFP.

Collectively, our data highlight the pervasive neofunctionalization among TFP components since seven paralogous gene sets exerted mainly opposing regulatory functions on movement and natural competence. For instance, paralogs of the minor pilins *pilEVWX* and *fimT* and the TFP tip adhesin *pilY1* are mostly organized as operons in two different regions of the *X. fastidiosa* TemeculaL genome (S2 Table). Here, we observed that individual mutations in one set of these paralogs abrogated twitching motility and greatly reduced natural competence, while deletions in the other set of paralogous genes produced variable phenotypes that were mostly opposite to their counterparts. It has been demonstrated that minor pilins prime the assembly of TFP [53]. This interaction is specific, in which major pilins require a specific cognate subset of minor pilins for TFP assembly [26]. Therefore, the presence of paralogs for major and minor pilins displaying opposite functions within *X. fastidiosa* possibly suggest assembly of multiple TFP fibers with specific functions. However, this needs to be further assessed (see SI discussion in S1 Text).

Most remarkably among paralogs, is the discovery of the minor pilin FimT3 as the TFP DNA receptor. This is an outstanding neofunctionalization example since FimT1 and FimT2 did not show DNA adherence abilities. This finding was also supported by our phylogenetic analysis, which showed divergent evolution of FimT3 from its paralogs. FimT3 plays an essential role in natural competence by mediating extracellular DNA uptake that may be integrated into the genome, thereby conferring new phenotypic traits that may allow an increase in bacterial fitness and/or host expansion of *X. fastidiosa*. The interaction between DNA molecules and an extracellular specific receptor is a key first step in the DNA uptake and thus natural competence of bacterial species. Initially, DNA binding by TFP has been observed in *P. aeruginosa* [66], *Streptococcus pneumoniae* [45], and *V. cholerae* [24], in which pilus retraction drives

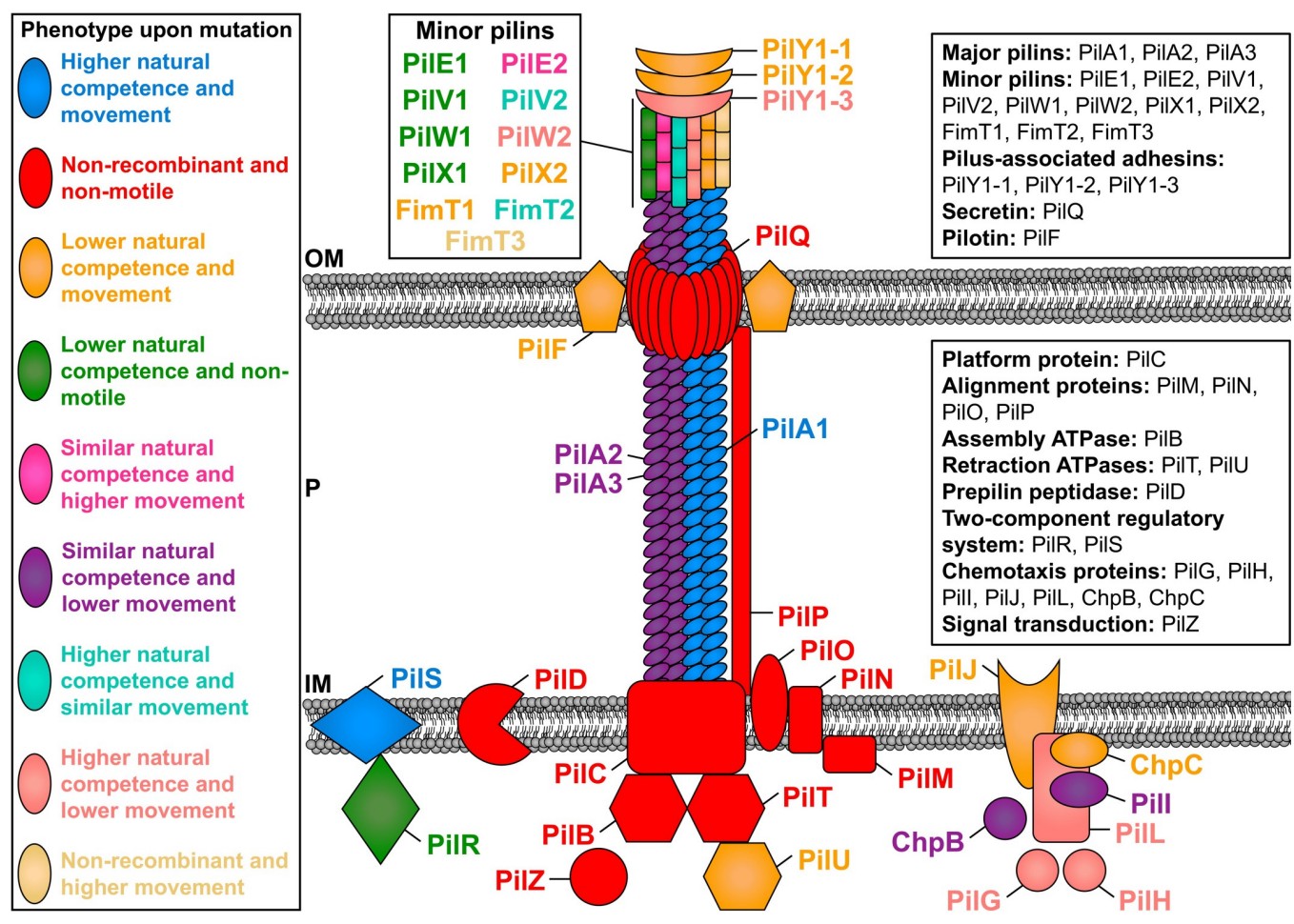

**Fig 5. Schematic representation of the functional role of TFP molecular components in natural competence and twitching motility of *X. fastidiosa*.** The figure shows the structure of a type IV pilus (PilA1, PilA2 and PilA3), the alignment subcomplex (PilM, PilN, PilO and PilP), the inner membrane motor subcomplex (PilB, PilC, PilD, PilT and PilU), the pore subcomplex (PilF and PilQ), minor pilins (PilE1, PilE2, PilV1, PilV2, PilW1, PilW2, PilX1, PilX2, FimT1, FimT2 and FimT3), TFP tip adhesins (PilY1-1, PilY1-2 and PilY1-3), as well as regulatory proteins (PilG, PilH, PilI, PilJ, PilL, PilR, PilS, PilZ, ChpB and ChpC). Proteins in the figure are color-coded according to their phenotype upon knockout mutation and thus functional role. More information about the role of each protein is described in S1 Table, throughout the manuscript and in SI Discussion in S1 Text. OM–outer membrane; P–periplasm; IM–inner membrane.

DNA uptake by transporting it to the cell surface, which is then imported to the periplasm by the competence protein ComEA [24, 70]. Although these early studies provide evidence that the specialized DNA receptor resides within type IV competence pilus fibers, no direct binding of DNA by individual purified major and/or minor pilins was reported. In *V. cholerae*, targeted mutations of positively charged amino acid residues within the minor pilins VC0858 and VC0859 reduced the DNA-binding affinity of its TFP [24], but the ability of these minor pilins to directly bind DNA was not assayed. Direct DNA binding has been first observed for the pilin protein ComP from *N. meningitidis*, with binding occurring at a positively charged surface of this protein [12]. ComP orthologs are found mostly within the Neisseriaceae family, where they mediate DNA uptake by binding to specific DNA uptake sequences, as *Neisseria* spp. preferentially take up homotypic DNA [6, 12]. ComZ, encoded by *T. thermophilus*, was the second minor pilin described as a TFP DNA receptor [61]. However, ComZ is an unusual minor pilin that possesses an additional large β-solenoid domain inserted into the common β-sheet structure of pilins. Furthermore, ComZ binds to another minor pilin, PilA2 (PilV in *X.*

*fastidiosa*), and its DNA binding ability does not appear to require positive charges in its surface [61]. Nonetheless, the binding of ComZ to another minor pilin suggests that specific interactions within the pilus structure may occur to assemble the competence pilus fiber.

Here, the deletion of individual paralogs of the major pilin PilA abrogated neither natural competence nor twitching motility, suggesting functional overlapping among them (see SI Discussion in S1 Text). In addition to the novel functions described above, our study confirmed the role of core components that were previously analyzed in other bacterial species including *Acinetobacter baumannii* [76], *A. baylyi* [46], *T. thermophilus* [60], *Haemophilus influenzae* [10], *P. aeruginosa* [1, 5, 49, 74], *L. pneumophila* [8] and *X. fastidiosa* [44, 47, 52]. The exceptions were *pilP* and *pilZ* in which previous deletion of both greatly reduced natural competence [22, 29], but did not abolish it here. It is of note that very few of these genes have been simultaneously assayed for their role in both natural transformation and movement mediated by TFP.

Deletion of FimT3 in *X. fastidiosa*, but not of the paralogs FimT1 and FimT2, exclusively disrupted the natural competence of this bacterium. FimT3 binds to DNA via a surface in which positively charged arginine residues contribute to the DNA binding ability of this protein. However, the binding appears to be non-specific, which is in contrast to ComP [12]. FimT from the human pathogen *L. pneumophila* [8], which is a homologue of the FimT3 studied here, was recently identified as a pilin with DNA-binding ability. Another FimT homolog was also demonstrated to be specialized in natural competence in *A. baylyi* [46], although no DNA binding was assessed. In our study, FimT3 seems to be particularly enriched in plant pathogens of the Xanthomonadaceae family, being encoded by all *X. fastidiosa* strains as well as many other bacterial species belonging to this family. Thus, our results, together with previous studies, demonstrate that DNA binding by TFP occurs among bacterial species through a similar component such as minor pilins, which themselves are not widely distributed or conserved, but are rather specific to different bacterial groups. For instance, the ability of strain-specific subsets of minor pilins to modulate TFP dynamics has already been described in *P. aeruginosa* (see SI Discussion in S1 Text) [26]. Understanding how this minor pilin interacts with different minor and major pilins of *X. fastidiosa* in the competence pilus is an important area to explore. It is not known whether the other Xanthomonadaceae bacterial species encoding FimT3 that were found in our analysis are also naturally competent. For instance, *X. fastidiosa* strains belonging to the *pauca* subspecies have a shorter FimT3 amino acid sequence, missing part of the C-terminal, and have a few aminoacidic changes (although maintain the arginine residues studied here). The only reported attempt to transform a *X. fastidiosa* strain belonging to the subspecies *pauca* by natural competence in vitro has failed [18]. Studies with the *X. fastidiosa* subsp. *pauca* FimT3 variant should investigate both individual DNA-binding activity and potential roles of the C-terminus in interaction with the rest of the TFP structure.

In summary, our study highlights the fundamental functions of TFP in virulence and evolution evidenced by complex regulatory features. Due to the worldwide threat that *X. fastidiosa* presents to many economically important crops and to the ever-increasing list of plant hosts, it is imperative to investigate how this plant pathogen acquires foreign DNA and expands its genetic diversity to help researchers understand its eco-evolutionary history and re-emergence of this pathogen in many geographic areas. In addition, our results provide a robust blueprint to assess TFP functions in bacteria living under flow conditions. An extensive discussion including additional genes analyzed in this study and their functional roles in natural transformation and twitching motility of *X. fastidiosa*, in addition to unique behaviors and other phenotypes such as biofilm formation, virulence in planta, and diversity in paralogous genes functions, is included in SI Discussion in S1 Text. All of this indicates that a complex and tightly regulated machinery is involved in these bacterial functions.

## Materials and methods

Methods that are only cited here are described in detail in SI Materials and Methods (in S1 Text).

### Bacterial strains, plasmids, and culture conditions

All strains and plasmids used in this study are listed in S6 Table. All *X. fastidiosa* mutant strains used here are derivatives of the *X. fastidiosa* subsp. *fastidiosa* strain TemeculaL [59], and were originated by site-directed mutagenesis of each gene of interest using natural competence as described elsewhere [40]. Primers used in this process are listed in S7 Table. *X. fastidiosa* was recovered from -80˚C glycerol stocks and routinely cultured for seven days at 28˚C on periwinkle wilt (PW) agar plates [19], modified by removing phenol red and using bovine serum albumin (1.8 g/l) (BSA; Gibco Life Sciences Technology), and sub-cultured onto fresh PW agar plates for another seven days at 28˚C before use. *X. fastidiosa* mutant strains were grown similarly but using PW plates amended with Km. All assays were performed using the subcultured *X. fastidiosa* strains. Cells were suspended and cultured in PD3 broth [19] to perform phenotypic assays, while phosphate-buffered saline (PBS) was used to suspend cells in liquid for in planta assays. Bacterial suspensions were performed by thorough pipetting followed by agitation using a vortex to foster greater homogenization and minimize the effects of cell-cell adhesion. Whenever needed, the antibiotics Km and Cm were used at concentrations of 50 and 10 μg/ml, respectively. When used together with Cm, the Km concentration was reduced to 30 μg/ml. Luria-Bertani (LB) medium (BD Difco) was used to culture *E. coli* cells. When needed, Km, Cm and ampicillin (Amp) were added to LB at concentrations of 50, 35 and 100 μg/ml, respectively.

### Phenotypic analyses of *X. fastidiosa* WT and mutant strains

Analyses of natural competence among strains [39], twitching motility [38], growth curve and growth rate [39], biofilm formation and planktonic growth [17, 39], settling rate [39], piliation observation by TEM [40], and virulence in planta using tobacco as model system [28] were performed as described elsewhere. All experiments had at least three biological replicates, unless otherwise stated.

### DNA uptake assays

DNA uptake assays were performed as similarly described for the analysis of natural competence among *X. fastidiosa* strains (see SI Materials and Methods in S1 Text). Briefly, recipient cells were suspended in PD3 broth to $OD_{600nm}$ = 0.6, spotted onto PD3 agar plates and grown for three days at 28˚C. Then, 1 μg of fluorescently labeled pAX1-Cm plasmid (10-μl volume), labeled using the *Label* IT Nucleic Acid Labeling kit, Cy3 (Mirus Bio LLC), was added on top of cells, air-dried, and incubated at 28˚C for another 24 h. After, cells were harvested in 150 μl of PD3 broth, and a 50 μl aliquot was treated with 10 units of DNase I (New England Biolabs) for 10 minutes at 37˚C to degrade the remaining extracellular DNA, and cells were observed using a 100× oil immersion objective in a Nikon Eclipse Ti inverted microscope (Nikon). Image acquisition was performed using a Nikon DS-Q1 digital camera (Nikon) controlled by the NIS-Elements software version 3.0 (Nikon), which was also used to create merged fluorescent images. To detect Cy3, an excitation wavelength of 590 nm was used (tetramethyl rhodamine isothiocyanate; TRITC filter). The proportion of cells that acquired extracellular DNA was calculated as the percentage of cells with fluorescent DNA foci to cells without fluorescent DNA foci. Experiments were performed at least three times independently.

## Protein cloning, expression, and purification

Cloning, expression, and purification of FimT1s, FimT2s and FimT3s were performed similarly as described elsewhere using the pHIS-Parallel1 plasmid (S6 Table) [71, 78]. Amino acid exchanges in FimT3s were performed by designing a pair of primers (S7 Table) that exchanged the arginine residues at positions 160 and 162 by alanine residues. This pair of primers was designed using the QuickChange Primer Design tool (Agilent Technologies, Inc.; https://www.agilent.com/store/primerDesignProgram.jsp). For amino acid exchanges, the whole pHIS-Parallel1-*fimT3s* construct was amplified via *Pfu* DNA polymerase (G-Biosciences) using a standard protocol from the manufacturer and this pair of primers in a S1000 thermal cycler (Bio-Rad). Expression and purification of FimT3s-R160AR162A were performed as described above (see SI Materials and Methods in S1 Text) [78]. PCR products were purified using the Gel/PCR DNA Fragments Extraction kit (IBI Scientific). Correct cloning and amino acid exchanges in all constructs were confirmed by Sanger sequencing (Sequetech Corporation) using the T7 promoter forward primer (S7 Table).

## DNA binding assays

Agarose EMSAs were mainly performed to assess the DNA-binding ability of purified FimT1s, FimT2s and FimT3s. Briefly, 200 ng of DNA (usually pAX1-Cm plasmid) were incubated for 30 minutes at 28°C with increasing concentrations of purified proteins in 20 μL EMSA reaction buffer (50 mM Tris-HCl, pH 7.5; 50 mM NaCl; 200 mM KCl; 5 mM MgCl$_2$; 5 mM EDTA, pH 8.0; 5 mM DTT; 0.25 mg/ml BSA). Lysis buffer containing 250 μM imidazole, in which proteins were suspended, was included as blank control. After the incubation period, DNA was separated by electrophoresis on a 0.8% agarose gel containing GelRed nucleic acid gel stain (Biotium) in Tris-acetate-EDTA buffer (120 V for 40 minutes). Experiments were performed at least three times independently. On the other hand, native acrylamide EMSAs were used to perform titration of the DNA binding activity of FimT3s and FimT3s-R160AR162A. In summary, 60 ng of Cy-3 labeled Km resistance cassette were incubated for 30 minutes at 28°C with increasing concentrations of purified proteins in 20 μL EMSA reaction buffer and separated by electrophoresis on a 3.5% native acrylamide gel in Tris-borate-EDTA buffer (40 V for 4 hours). Native acrylamide gels were pre-run at 40 V for 30 minutes before being used. DNA samples were directly visualized using the ImageQuant LAS 4010 Imaging System (GE Healthcare), since the used DNA was fluorescently labeled with Cy3. Experiments were performed two times independently. Densitometry analysis of the shifted bands of Cy-3 labeled DNA in native acrylamide gels was performed by measuring the fluorescence intensity of these bands when treated with 1, 2, 5, and 10 μM of each protein using ImageJ [67]. Then, these values were used to calculate the area under the fluorescence curves, as performed to obtain the AUDPC (see SI Materials and Methods in S1 Text) [72], to quantify the DNA-binding affinity of FimT3s and FimT3s-R160AR162A by determining the progress of shifted bands in relation to each protein concentration.

## Bioinformatic analyses

FimT3 modeling was performed using the Phyre2 web portal [42] and visualized via PyMOL version 2.4.0 (Schrödinger, LLC). The surface electrostatics of FimT3 was predicted using the APBS Electrostatics plugin from PyMOL. Sequence alignments of FimT3 with ComP, VC0858 and FimT from *A. baylyi* and *L. pneumophila* were performed by retrieving the respective sequences from NCBI, aligning through the T-Coffee Multiple Sequence Alignment Server [21] and visualizing using the BoxShade webserver (https://embnet.vital-it.ch/software/BOX_form.html). For FimT3 analyses within the Xanthomonadaceae family, DNA sequences of

different FimT homologs from *Xylella*, *Xanthomonas*, *Lysobacter* and *Stenotrophomonas* were used as query for screening genomes of Xanthomonadaceae using CRB-BLAST [4] with default settings. Nucleotide sequences for the CRB-BLAST hits were then retrieved and aligned using MAFFT [41], followed by RAxML version 8.0.24 [73] to build a phylogenetic tree, which was visualized using Fig Tree version 1.4.4 (http://tree.bio.ed.ac.uk/). FimT3-encoding sequences were retrieved from the phylogenetic tree using the TREE2FASTA Perl script [63], translated into amino acid sequences and aligned using Clustal Omega (Clustal 12.1) to determine percentage of identical amino acids [48]. The visualization of the alignment of representative FimT3 sequences among *X. fastidiosa* strains and representative bacterial species belonging to the Xanthomonadaceae family was performed using the BoxShade webserver. The FimT3 GRxR motif sequence logo was generated using the WebLogo webserver [16].

## Data analysis

Data from natural competence assays (recombination frequency) and the area under the fluorescence curve were compared by two-tailed Student's *t*-test. Data from twitching motility, growth rate, viable *X. fastidiosa* CFU/ml obtained during natural competence assays, biofilm formation, planktonic growth, settling rate, AUDPC, *X. fastidiosa* population in planta and percentage of cells acquiring DNA from the extracellular environment were individually analyzed by one-way analysis of variance (ANOVA) followed by Tukey's HSD multiple comparisons of means in R 4.0.0 under the package multcomp [31]. Correlation among analyzed phenotypes was determined by Pearson's correlation using the SigmaPlot software version 11.0 (Systat Software Inc.).

## Supporting information

**S1 Text. Includes expanded Materials and Methods and an in-depth Discussion of the role of each TFP gene based on our phenotypic characterization.**
(DOCX)

**S1 Table. List of *pil* and associated genes deleted in *X. fastidiosa* strain TemeculaL.**
(PDF)

**S2 Table. Genomic organization of *pil* and associated genes in *X. fastidiosa* TemeculaL.**
(PDF)

**S3 Table. Pearson's correlation results for the analyzed phenotypic traits of *X. fastidiosa*.**
(PDF)

**S4 Table. DNA-binding probability of amino acid residues from FimT3.**
(PDF)

**S5 Table. Homologs of FimT3 are found in members of the Xanthomonadaceae family.**
(PDF)

**S6 Table. Bacterial strains and plasmids used in this study.**
(PDF)

**S7 Table. List of PCR primers and qPCR primers and probe used in this study.**
(PDF)

**S1 Fig. Recombination frequencies obtained through natural transformation of *X. fastidiosa* mutant strains used in this study.** Quantification of natural transformation was performed in PD3 plates by applying 1 μg of the pAX1-Cm plasmid to equivalent numbers of

recipient cells of each *X. fastidiosa* strain to generate chloramphenicol-resistant (Cm$^R$) mutations. Total viable cells and transformants were counted and results are expressed as the ratio of recipient cells transformed. WT is highlighted in blue, and the dashed blue line indicates the mean value of recombination frequency for the WT. Data represent means and standard errors. $^*$ and $^{**}$ indicate significant difference ($P<0.05$ and $P<0.005$, respectively) of recombination frequency through natural competence in comparison to the WT as determined using Student's *t*-test (n = three to 21 independent replicates with two internal replicates each). "$<$" indicates below detection limit, which was $10^{-7}$. Mutant strains below the detection limit were considered non-recombinant.
(TIF)

**S2 Fig. Fringe width measurements of the twitching motility phenotypes of *X. fastidiosa* mutant strains used in this study.** Twitching motility was determined by spotting cells of each strain in PW without BSA plates and measuring the movement fringe width after 4 days of growth at 28°C. WT is highlighted in blue, and the dashed blue line indicates the mean value of fringe width for the WT. Data represent means and standard errors. Different letters on top of bars indicate significant difference as analyzed by ANOVA followed by Tukey's HSD multiple comparisons of means ($P<0.05$; n = three to 14 independent replicates with eight to 48 internal replicates each). The detection limit was 10 μm. Mutant strains below the detection limit were considered non-motile.
(TIF)

**S3 Fig. Figure panel with representative pictures of the twitching motility phenotypes of *X. fastidiosa* mutant strains used in this study.** The assay was performed as described in S2 Fig. Similar events were captured in three to 14 independent experiments. Images were captured at 10× magnification. Scale bar (right lower panel), 100 μm.
(PDF)

**S4 Fig. Growth curves of *X. fastidiosa* mutant strains used in this study.** Growth curves were generated by culturing bacteria in PD3 broth within 96-well plates and measuring the optical density at 600 nm (OD$_{600nm}$) values each day for 8 days. Data represent means and standard errors (n = three to 15 independent replicates, with eight internal replicates each).
(TIF)

**S5 Fig. Growth rates of *X. fastidiosa* mutant strains used in this study.** Growth rate was calculated from the growth curve at the exponential growth phase (2 to 6 days post inoculation). WT is highlighted in blue, and the dashed blue line indicates the mean value of growth rate for the WT. Data represent means and standard errors. Different letters on top of bars indicate significant difference as analyzed by ANOVA followed by Tukey's HSD multiple comparisons of means ($P<0.05$; n = three to 15 independent replicates with eight internal replicates each).
(TIF)

**S6 Fig. Total number of viable CFU/ml of *X. fastidiosa* mutant strains used in this study after growth during natural competence assays.** Total viable CFUs of each *X. fastidiosa* strain obtained during natural competence assays described in S1 Fig are shown here. WT is highlighted in blue, and the dashed blue line indicates the mean value of the total number of viable CFU/ml obtained during natural competence assays for the WT. Data represent means and standard errors. Different letters on top of bars indicate significant difference as analyzed by ANOVA followed by Tukey's HSD multiple comparisons of means ($P<0.05$; n = three to 21 independent replicates with two internal replicates each).
(TIF)

**S7 Fig. Biofilm formation of *X. fastidiosa* mutant strains used in this study.** Biofilm was measured by staining *X. fastidiosa* cells attached to the 96-well plates at the end of the growth curve experiment with crystal violet. WT is highlighted in blue, and the dashed blue line indicates the mean value of biofilm formation for the WT. Data represent means and standard errors. Different letters on top of bars indicate significant difference as analyzed by ANOVA followed by Tukey's HSD multiple comparisons of means ($P<0.05$; n = three to 14 independent replicates with six to eight internal replicates each).
(TIF)

**S8 Fig. Planktonic growth of *X. fastidiosa* mutant strains used in this study.** Planktonic growth was quantified by measuring the optical density at 600 nm ($OD_{600nm}$) values of the supernatant of each *X. fastidiosa* strain at the end of the growth curve experiment. WT is highlighted in blue, and the dashed blue line indicates the mean value of planktonic growth for the WT. Data represent means and standard errors. Different letters on top of bars indicate significant difference as analyzed by ANOVA followed by Tukey's HSD multiple comparisons of means ($P<0.05$; n = three to 14 independent replicates with six to eight internal replicates each).
(TIF)

**S9 Fig. Settling rates of *X. fastidiosa* mutant strains used in this study.** Settling rate was measured by suspending ($OD_{600nm} = 1.0$) the analyzed *X. fastidiosa* strains in a cuvette in 1 ml of PD3 broth and measuring $OD_{600nm}$ values at the initial time point and after 2 hours. WT is highlighted in blue, and the dashed blue line indicates the mean value of settling rate for the WT. Data represent means and standard errors. Different letters on top of bars indicate significant difference as analyzed by ANOVA followed by Tukey's HSD multiple comparisons of means ($P<0.05$; n = three to 15 independent replicates).
(TIF)

**S10 Fig. *X. fastidiosa* cells take up Cy-3 labeled DNA into a DNase I resistant state.** WT and mutant cells were exposed to Cy-3-labeled pAX1-Cm plasmid (1 μg) for 24 hours, treated with DNase I and DNA foci were observed using a fluorescence microscope. The images shown correspond to the bright field (left), Cy-3 channel (center) and merged fluorescent images (right). In merged fluorescent images, arrows are pointing to fluorescent DNA foci at the poles of cells. Similar events were captured in two to three independent experiments. All evaluated cells presented uptake of Cy-3 labeled DNA. Mutant strains analyzed here correspond to knockout of minor pilins that presented lower recombination through natural competence in comparison to the WT. Δ*fimT1* and Δ*fimT2* were included for comparison with Δ*fimT3*. Images were captured at 100× magnification. Scale bars, 1.5 μm.
(TIF)

**S11 Fig. Percentage of *X. fastidiosa* cells that acquired DNA from the extracellular environment during DNA uptake assays in the different analyzed strains.** Total cells and cells with DNA foci (Cy-3 labeled pAX1-Cm plasmid) were counted and results are expressed as the percentage of cells with DNA foci. All mutants of minor pilins apart Δ*pilX2* and Δ*fimT2* presented significant lower DNA uptake than WT cells. Data represent means and standard errors. Different letters on top of bars indicate significant difference as analyzed by ANOVA followed by Tukey's HSD multiple comparisons of means ($P<0.05$; n = two to three independent replicates with three to seven technical replicates each).
(TIF)

**S12 Fig. FimT3 DNA binding activity in different DNA sequences.** DNA-binding activity of purified FimT3s (5 μM) to different DNA sequences was assessed by agarose EMSA using standard amounts (200 ng) of each DNA sequence. FimT3s was incubated with each DNA sequence for 30 min at 28˚C and resolved by electrophoresis on a 0.8% agarose gel. Lysis buffer containing 250 μM imidazole, in which proteins were suspended, was included as blank control. DNA binding activity of FimT3s in large sequences (larger than 1,200 bp) is observed as clear shifts in the electrophoretic mobility of bands, while DNA binding activity of FimT3s in small sequences (smaller than 900 bp) is observed as smearing of bands. Amplicon sizes: pAX1-Cm– 4,361 bp; pGEM-T– 3,000 bp; pHIS-Parallel1-*fimT3s* – 5,977 bp; pLas16S – 5,206 bp; Km– 1,202 bp; *pilY1-2*-UP– 988 bp; *fimT3*–609 bp; *fimT3s* – 519 bp; *pilA3*-UP– 852 bp; *pilA3*-Down– 887 bp; *pilR*-UP– 900 bp; *pilR*-Down– 937 bp. The experiment included DNA sequences with homology to the genome of *X. fastidiosa* TemeculaL (pAX1-Cm, pHIS-Parallel1-*fimT3s*, *pilY1-2*-UP, *fimT3*, *fimT3s*, *pilA3*-UP, *pilA3*-Down, *pilR*-UP and *pilR*-Down), as well as sequences with no apparent homology (pGEM-T, pLas16S and Km resistance cassette). FimT3s did not appear to preferably bind certain DNA sequences, indicating that its DNA-binding ability is non-sequence-specific. *fimT3* –full-length *fimT3* amplified from *X. fastidiosa* TemeculaL; *fimT3s* –soluble portion of *fimT3* amplified from *X. fastidiosa* TemeculaL; sequences labeled with "UP" and "Down" correspond to upstream and downstream sequences, respectively, used to construct the targeting sequences for site-directed mutagenesis of each gene of interest. Similar events were captured in two independent experiments.
(TIF)

**S13 Fig. Alignment of FimT3 from *X. fastidiosa* to DNA-binding minor pilins from *Neisseria meningitidis* (ComP), *Vibrio cholerae* (VC0858), *Acinetobacter baylyi* (FimT) and *Legionella pneumophila* (FimT).** The sequence of each protein was obtained from NCBI, aligned through T-Coffee, and visualized using BoxShade. Black shading indicates conserved residues; grey shading indicates conservative mutations; and white color indicates divergence among sequences. **a**, **b** and **c** show alignment of FimT3 with ComP, VC0858, and FimT from *A. baylyi* and *L. pneumophila*, respectively. The arginine residue at position 162 (R162) from FimT3 aligned with a lysine of ComP (K108) demonstrated to be essential for the DNA-binding ability of the latter protein. On the other hand, the arginine residue at position 160 (R160) from FimT3 aligned with an arginine of VC0858 (R168) demonstrated to be important for the DNA-binding ability of the *V. cholerae* pilus. In addition, R160 aligned with an arginine residue of FimT from *A. baylyi*, and both R160 and R162 aligned with arginine residues of FimT from *L. pneumophila* that are important to its DNA-binding ability. Alignment of these specific amino acid residues are highlighted by a yellow shading in **a**, **b** and **c**.
(TIF)

**S14 Fig. The DNA-binding residues of FimT3 is highly conserved within bacterial members of the Xanthomonadaceae family encoding this protein.** The sequence of each protein was downloaded from NCBI, screened for the presence of FimT3 by tblastn, aligned through MAFFT, and visualized using BoxShade. Black shading indicates conserved residues; grey shading indicates conservative mutations; and white color indicates divergence among sequences. **a** Alignment of different FimT3 sequences among *X. fastidiosa* strains. FimT3 is nearly identical in all *X. fastidiosa* strains, with strains from subspecies *pauca* (strains 9a5c and Hib4) presenting the highest divergence (97.25% and 95.6% of identical amino acids, respectively, in comparison to FimT3 from strain TemeculaL). **b** Alignment of representative FimT3 sequences from bacterial members of the Xanthomonadaceae family encoding this protein. Although the percentage of identical amino acids ranged from 15.5% to 100%, most sequences aligned with the arginine amino acid residues at positions 160 and 162 of FimT3 from *X.*

*fastidiosa* strain TemeculaL (highlighted with a yellow shading in the figure). This indicates that these arginine residues are highly conserved within FimT3 sequences. For conciseness of the figure, only representative FimT3 sequences are shown in the alignment. The different portions of FimT3 (prepilin leader peptide, transmembrane domain, and soluble portion) are indicated in the figure. A sequence logo generated using the full alignment of 1,416 FimT3 sequences to highlight the conserved GRxR motif is shown in the bottom of the figure. Abbreviations are the following. ***Pseudoxanthomonas***: Pdo–*P. dokdonensis*; Pkl–*P. kalamensis*; Pme–*P. mexicana*; Psp–*P.* sp. ***Xanthomonas***: Xar–*X. arboricola*; Xcm–*X. campestris*; Xct–*X. citri*; Xme–*X. melonis*; Xsa–*X. sacchari*; Xso–*X. sontii*; Xsp–*X.* sp.; Xth–*X. theicola*; Xtr–*X. translucens*. ***Xylella***: Xf–*X. fastidiosa*.
(TIF)

## Acknowledgments

We thank Dr. Michelle Mendonça Pena for helping with phylogenetic analyses. We thank the Alabama Supercomputing Authority for granting access to their high-performance computing platform.

## Author Contributions

**Conceptualization:** Marcus V. Merfa, Neha Potnis, Paul A. Cobine, Leonardo De La Fuente.

**Data curation:** Marcus V. Merfa, Neha Potnis, Leonardo De La Fuente.

**Formal analysis:** Marcus V. Merfa, Xinyu Zhu, Deepak Shantharaj, Neha Potnis, Paul A. Cobine.

**Funding acquisition:** Leonardo De La Fuente.

**Investigation:** Marcus V. Merfa, Xinyu Zhu, Deepak Shantharaj, Eber Naranjo, Paul A. Cobine, Leonardo De La Fuente.

**Methodology:** Marcus V. Merfa, Xinyu Zhu, Deepak Shantharaj, Laura M. Gomez, Eber Naranjo, Neha Potnis, Paul A. Cobine.

**Project administration:** Leonardo De La Fuente.

**Resources:** Leonardo De La Fuente.

**Supervision:** Leonardo De La Fuente.

**Validation:** Marcus V. Merfa.

**Writing – original draft:** Marcus V. Merfa, Paul A. Cobine, Leonardo De La Fuente.

**Writing – review & editing:** Marcus V. Merfa, Xinyu Zhu, Deepak Shantharaj, Neha Potnis, Paul A. Cobine, Leonardo De La Fuente.

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
