## [Decision Letter · Decision Letter 0]

4 Jan 2023

Dear Dr De La Fuente,

Thank you very much for submitting your manuscript "Complete functional analysis of type IV pilus components of a reemergent plant pathogen reveals neofunctionalization of paralog genes" for consideration at PLOS Pathogens. As with all papers reviewed by the journal, your manuscript was reviewed by members of the editorial board and by several independent reviewers. The reviewers appreciated the attention to an important topic and were overall impressed by both the quality and presentation of the data. Based on the reviews, we are likely to accept this manuscript for publication, provided that you consider the minor suggestions of reviewers #1 and #3 and revise the manuscript accordingly.

Sincerely,

David Mackey

Academic Editor

PLOS Pathogens

Shou-Wei Ding

Section Editor

PLOS Pathogens

Kasturi Haldar

Editor-in-Chief

PLOS Pathogens

orcid.org/0000-0001-5065-158X

Michael Malim

Editor-in-Chief

PLOS Pathogens

orcid.org/0000-0002-7699-2064

Reviewer Comments (if any, and for reference):

Reviewer's Responses to Questions

**Part I - Summary**

Reviewer #1: Type IV pilus(TFP) involve in twitching motility, adhesion and even DNA exchange. The manuscript functional analyzed the TFP-relative genes in Xylella fastidiosa through detecting twitching motility, bacterial growth, virulence and also the natural transformation of a serious of mutant strains. Generally, the authors had done many experiments to identify TFP-relative gene function. And the work is interesting to reader and relative researchers.

Minor suggestions:

1. In Fig 1, the authors displayed the fringe width and recombination frequency of mutant strains in a combined figure. It is much clearer and more readable for reader to divide natural transformation and motility phenotype into Fig 1A and Fig 1B.

2. For the DNA-binding assay, why are there always a band locating at the protein size even at 0 uM protein in Fig. 2C and 2E?

3. For statistic analysis, the author described in Line 705, 809, 818, 829, 845, 852, 859 and 867 as “n = three to ** independent replicates with two internal replicates each”. It is more incredible or reasonable to give the detail number of replicates for each experiment.

4. Both minor pilins encoding gene pilE1 and fimT3 are required for the natural competence of acquiring DNA from extracellular environment (Fig S13), can PilE1 bind different DNA sequences similar as FimT3?

5. In Fig S10C and S10D, no significant difference of bacterial population of WT and mutant strains was found in top leaf. However, the symptoms caused by inoculation with different mutant strains displayed the clear differences.

Reviewer #2: This study demonstrated the roles of the 38 genes of Type IV pilus (TFP) of the Xylella fastidiosa in movement, natural competence and virulence The authors showed that ten genes were essential for both movement and natural competence, but seven sets of paralogs showed opposing phenotypes, indicating evolutionary neofunctionalization of subunits within TFP. The authors determined that the minor pilin FimT3 is the DNA receptor in TFP, and it is conserved among X. fastidiosa strains and can bind DNA in a non-specifically pattern. Two arginine residues in FimT3 are essential for DNA binding. I think the work is very important and valuable, also provided some insights in T4P function and also have referential value for other bacteria such as Xanthomonas oryzae pv. oryzae living under flow conditions.

Reviewer #3: This study has reported on the results of a rather comprehensive study of the various roles of type IV pili (TFP) in the important plant pathogenic bacterium Xylella fastidiosa. This study follows extensive prior work by this group and by others that have zeroed in on the important role of TFP in this pathogen both because of the important roles of TFP in attachment to surfaces, and in active movement of the pathogen in xylem vessels subject to high velocity liquid flow. This group also has contributed nearly everything that we know about the role of competency in this species, which complements the extensive genomic sequencing studies that have revealed a particularly large role of recombination and horizontal gene transfer in this species. This current study therefore is highly justified as it would be expected that TFP would play a role both in active motility and DNA uptake. What makes a study particularly impressive is the comprehensive nature by which the authors have identified and individually knocked out each of the many genes involved in TFP production and made functional assessments of the mutants. The study is particularly impressive because of the quantitative nature of their measurements of the contribution of each of these genes to both competency, as measured by recombination frequency and active motility, measured by quantitative twitching assays. The data presented in Figure 1 is this a real tour de force. While I found this aspect of this study to be the most biologically interesting and compelling, the authors also have presented reasonable data to support the conclusions that the minor pilus FimT3 is THE DNA receptor in TFP of X. fastidiosa. While I don't think there is any real argument that FimT3 is in fact the DNA receptor, I felt that they were straining a bit too try to demonstrate the essentiality of the two arginines in locations 160 and 162 in the data presented in lines 255 to 270. They seem to agree that the data and support of these two residues are somewhat soft, as represented in the very modest defects of alanine substitutions on gel shift studies shown in Figure 2. While the data is rather soft because of the very small effect of these substitutions, I don't have any argument in retaining it in the study. Perhaps I am simply not as interested in the phylogenetic components of the study, but the data addressing the divergent and conservation of fimT3 in X. fastidiosa, other Xanthomonadacea, and more broadly in bacteria didn't seem to really add too much to the story, although it did seem to explain why subspecies pauca is not competent. The arguments for neofunctionalization of these various TFP components is compelling. Their study does provide fodder for some interesting follow-on studies. Specifically, their finding that the presence of paralogs for major and minor pilins having opposite functions seem to suggest that there could be assembly of multiple TFP fibers having specific functions. I note that the manuscript is exceptionally well written. This is a refreshing change from the many manuscripts that I get to review that are very difficult to read.

**Part II – Major Issues: Key Experiments Required for Acceptance**

Reviewer #1: (No Response)

Reviewer #2: There are no comments on the experiments.

Reviewer #3: none needed

**Part III – Minor Issues: Editorial and Data Presentation Modifications**

Reviewer #1: Type IV pilus(TFP) involve in twitching motility, adhesion and even DNA exchange. The manuscript functional analyzed the TFP-relative genes in Xylella fastidiosa through detecting twitching motility, bacterial growth, virulence and also the natural transformation of a serious of mutant strains. Generally, the authors had done many experiments to identify TFP-relative gene function. And the work is interesting to reader and relative researchers.

Minor suggestions:

1. In Fig 1, the authors displayed the fringe width and recombination frequency of mutant strains in a combined figure. It is much clearer and more readable for reader to divide natural transformation and motility phenotype into Fig 1A and Fig 1B.

2. For the DNA-binding assay, why are there always a band locating at the protein size even at 0 uM protein in Fig. 2C and 2E?

3. For statistic analysis, the author described in Line 705, 809, 818, 829, 845, 852, 859 and 867 as “n = three to ** independent replicates with two internal replicates each”. It is more incredible or reasonable to give the detail number of replicates for each experiment.

4. Both minor pilins encoding gene pilE1 and fimT3 are required for the natural competence of acquiring DNA from extracellular environment (Fig S13), can PilE1 bind different DNA sequences similar as FimT3?

5. In Fig S10C and S10D, no significant difference of bacterial population of WT and mutant strains was found in top leaf. However, the symptoms caused by inoculation with different mutant strains displayed the clear differences.

Reviewer #2: Just there are a little advice for authors some results such as Result 1 was written too complicated

Reviewer #3: Specifics:

Lines 49 through 50. The sentence seems to accurately summarize the results of this study, but if they have the space, I think it would have more impact if there was some a statement about whether the presence of fimT3 was linked to competency in any of these other taxa.

Line 141. The sentence is a bit awkward in that they should indicate that deletion of pilR showed that PilR was essential for twitching mobility but not transformation.

Lines 173 through 175. The authors have not provided enough information in the materials and methods for me to evaluate whether they have indeed overcome the complications of the likely clumping of cells in these assays. X. fastidiosa is notorious for cell-cell adhesion. The authors clearly understand that growth rate measurements done by measuring optical density is prone to issues, but it is not clear how their simple measurement of CFU/milliliter by dilution plating overcomes this issue unless there was an attempt to disperse the cells by sonication or some other method to disassociate the cells. CFU/ml can differ hugely from cells/ml, which is their goal here, if in fact the cells have not been dispersed.

Lines 189 through 206. I was disappointed and surprised that they did not place their data on the virulence measurements of these strains in a figure in the text rather than as a supplemental figure. I think a lot of readers would like to see this type of information, and many would be reluctant to dig into the supplement to find it. Since they only have a small number of figures already, I think it would be better to put it in the main text.

Lines 215-217. The sentence is awkward. It would seem that they have used DNase1 to remove any extracellular DNA, but this sentence does not make that clear.

Line 227. ... and compare it to its paralogs..

Line 280 ... of the Xanthomonadaceae ( 47%), the majority...

PLOS authors have the option to publish the peer review history of their article (what does this mean?). If published, this will include your full peer review and any attached files.

Reviewer #1: No

Reviewer #2: No

Reviewer #3: **Yes: **Steven Lindow

Figure Files:

Data Requirements:

Reproducibility:

References:

---

## [Editor Report · Decision Letter 1]

26 Jan 2023

Dear Dr De La Fuente,

We are pleased to inform you that your manuscript 'Complete functional analysis of type IV pilus components of a reemergent plant pathogen reveals neofunctionalization of paralog genes' has been provisionally accepted for publication in PLOS Pathogens.

Best regards,

David Mackey

Academic Editor

PLOS Pathogens

Shou-Wei Ding

Section Editor

PLOS Pathogens

Kasturi Haldar

Editor-in-Chief

PLOS Pathogens

orcid.org/0000-0001-5065-158X

Michael Malim

Editor-in-Chief

PLOS Pathogens

orcid.org/0000-0002-7699-2064
---

## [Editor Report · Acceptance letter]

8 Feb 2023

Dear Dr De La Fuente,

We are delighted to inform you that your manuscript, "Complete functional analysis of type IV pilus components of a reemergent plant pathogen reveals neofunctionalization of paralog genes," has been formally accepted for publication in PLOS Pathogens.

Best regards,

Kasturi Haldar

Editor-in-Chief

PLOS Pathogens

orcid.org/0000-0001-5065-158X

Michael Malim

Editor-in-Chief

PLOS Pathogens

orcid.org/0000-0002-7699-2064